# SlotFormer: Long-Term Dynamic Modeling in Object-Centric Models

**Ziyi Wu**[1,2]     **Nikita Dvornik**[3,1]     **Klaus Greff**[4]     **Jiaqi Xi**[5]     **Thomas Kipf**[*4]     **Animesh Garg**[*1,2]

[1]University of Toronto
[2]Vector Institute
[3]Samsung AI Centre Toronto
[4]Google Research
[5]Peking University

## Abstract

Understanding dynamics from visual observations is a challenging problem that requires disentangling individual objects from the scene and learning their interactions. While recent object-centric models can successfully decompose a scene into objects, modelling their dynamics effectively still remains a challenge. We address this problem by introducing SlotFormer - a Transformer-based autoregressive model operating on learned object-centric representations. Given a video clip, our approach performs dynamic reasoning over object features to model spatial-temporal object relationships and generate realistic future frames. In this paper, we successfully apply SlotFormer to the problem of consistent long-term dynamic modeling in object-centric models. We compare Slot-Former to image-based video prediction models and object-centric dynamic models on two synthetic video datasets consisting of complex object interactions. Our method generates videos of high quality as measured by conventional video prediction metrics, while achieving significantly better long-term synthesis of object dynamics.

## 1 INTRODUCTION

Visual reasoning in videos often involves object-oriented scene decomposition [Yi et al., 2019] and temporal dynamics understanding [Chen et al., 2020b, Ding et al., 2021b]. A traditional approach to video prediction [Shi et al., 2015, Wang et al., 2017, 2018b, Denton and Fergus, 2018, Yu et al., 2019] relies on global dense frame-level representations, which uses past frames feature maps to predict future frames representations. Such models are lacking object-specific inductive biases and often produce blurry generation outputs, failing to preserve object identities over time [Lotter et al.,

2016, Wang et al., 2018a]. Fortunately, this issue can be mitigated if using recently proposed object-centric models [Burgess et al., 2019, Greff et al., 2019, Locatello et al., 2020]. This class of methods first constructs a structured representation of the scene, and then learns the interactions among these object-centric features (a.k.a. slots) to model future object dynamics [Jiang et al., 2019, Kossen et al., 2019, Lin et al., 2020].

While the general direction is promising, most existing works bake in strong assumptions in their frameworks. This includes restrictive domain-specific scene priors [Jiang et al., 2019] or hand-crafted properties for object representations [Lin et al., 2020]. As a result, such meticulously engineered methods are successful on simple 2D datasets, yet failing to scale to more complex 3D environments [Yi et al., 2019, Hill et al., 2020]. Alternatively, more generic methods, such as [Creswell et al., 2021] and [Zoran et al., 2021], can be applied to a wider range of domains, but struggle to model object dynamics faithfully. This is expected, as such methods model object interactions and their temporal dynamics separately (using a weaker sequence model, i.e., an LSTM [Hochreiter and Schmidhuber, 1997], for the latter). In addition, all the previous works only experiment on datasets with limited object movement and relatively short rollout horizon (often smaller than 20 steps).

In this paper, we address the aforementioned shortcomings with SlotFormer: a purely Transformer-based autoregressive model. We treat future prediction as a sequential modeling problem: given a sequence of input images, SlotFormer takes in the object-centric representations extracted from these frames, and predicts the object features in the future step. Importantly, as opposed to many previous works, we do not inject any priors to the model; both the object-centric and the dynamic model are generic learnable modules. Besides, by conditioning on multiple frames, the self-attention operation in Transformers is capable of capturing the spatial-temporal object interactions simultaneously. It ensures the consistency of object properties and dynamics in the synthesized frames. We evaluate SlotFormer on two video datasets

*Accepted for the Causal Representation Learning workshop at the 38th Conference on Uncertainty in Artificial Intelligence* (UAI CRL 2022).

consisting of diverse object dynamics. Our method not only presents competitive results on standard video prediction metrics, but also achieves significant gains when evaluating on object-aware metrics in the long range. In summary, our main contributions are:

- A Transformer-based dynamic model for future synthesis in object-centric models.
- State-of-the-art performance on OBJ3D and CLEVERER, two datasets with complex object interactions, where our approach outperforms the baselines by a sizeable margin in modeling long-term dynamics.

## 2 RELATED WORK

In this section, we provide a brief overview of related works on object-centric modeling and Transformers, which is further expanded in Appendix A.

**Object-centric representation learning from videos.** Our work builds upon recent effort in decomposing raw videos into temporally aligned *slots* [Crawford and Pineau, 2020, Kipf et al., 2021, Kabra et al., 2021]. Existing object-centric dynamic models often make strong assumptions on the underlying object representations. Jiang et al. [2019] explicitly decompose the scene into foreground and background to apply fixed object size and presence priors. Lin et al. [2020] further disentangle object features to represent object positions, depth and semantic attributes separately. Some works leverage the power of Transformers to eliminate these domain-specific priors, while they still model the single-step object interactions and temporal scene dynamics separately [Creswell et al., 2021, Zoran et al., 2021]. The most relevant work to ours is OCVT [Wu et al., 2021], which also applies Transformers to slots from multiple frames. However, OCVT utilizes manually disentangled object features, and needs Hungarian matching for latent alignment during training. Therefore, it still underperforms RNN-based baselines in the future prediction task. In contrast, SlotFormer is a general Transformer-based dynamic model which is agnostic to the underlying object-centric representations. It performs spatial-temporal reasoning over objects simultaneously, enabling consistent long-term dynamics modeling.

**Transformers for sequential modeling.** Inspired by the success of autoregressive Transformers in language modeling [Radford et al., 2018, 2019, Brown et al., 2020], they are also adapted to several image and video generation tasks [Esser et al., 2021, Yan et al., 2021, Rombach et al., 2021, Ren and Wang, 2022]. To handle the high dimensionality of images, these models often adopt a two-stage training strategy by first mapping images to discrete tokens Chen et al. [2020a], Esser et al. [2021], and then learning a model over tokens. However, since they operate on a regular image grid, their mapping ignores the boundary of objects and usually splits one object into multiple tokens. In this work, we learn

a transformer-based dynamics model over *slot*-based representations that capture the entire object in a single vector thus showing high-quality results.

## 3 METHOD

In this section, we describe our Transformer-based autoregressive model for dynamics modeling. We first review the object-centric models we build upon (Section 3.1), then, present the SlotFormer model (Section 3.2), and, finally, describe the training objectives of our method (Section 3.3).

### 3.1 REVISITING SAVI

SlotFormer can build on any object-centric model that is able to decompose video frames into temporally-aligned object slots. We employ SAVi [Kipf et al., 2021] as our base model in this paper due to its strong performance in unsupervised object discovery and efficient inference process.

Given a series of input frames $\{x_t\}_{t=1}^T$, SAVi first applies a Convolutional Neural Network (CNN) encoder to extract image features, adds positional encoding and flattens the result, $h_t = f_{enc}(x_t) + p_{pos} \in \mathbb{R}^{(HW) \times D_{enc}}$. Then, the model initializes $N$ slots $\tilde{S}_1 \in \mathbb{R}^{N \times D_{slot}}$ from a set of learnable vectors, and performs Slot Attention [Locatello et al., 2020] between slots and the visual features. This process is repeated at every timestep, denoted as:

$$ S_t = f_{SA}(\tilde{S}_t, h_t). \tag{1} $$

The iterative attention updates the slot representations to capture individual objects, thus decomposing the image. The slot initialization for the next time-step is obtained from the processed slots in the previous time-step as follows:

$$ \tilde{S}_{t+1} = f_{dyn}(S_t), \tag{2} $$

where $f_{dyn}$ is a transformer encoder. That is, by alternating between Eq. (1) and (2), SAVI decmoposes a video into a set of temporarily consistent slots.

Finally, SAVi uses Spatial Broadcast Decoder [Watters et al., 2019] to decode each slot into an RGB image $y_t^n$ and a segmentation mask $m_t^n$ (segmenting the slots's object), which are combined into the final reconstructed image $\hat{x}_t$:

$$ (m_t^n, y_t^n) = f_{dec}(s_t^n), \quad \hat{x}_t = \sum_{n=1}^N m_t^n \odot y_t^n. \tag{3} $$

The entire network is trained end-to-end using a Mean Squared Error (MSE) loss between $x_t$ and $\hat{x}_t$.

One natural way of extending SAVi to future prediction is enforcing $\hat{S}_t$ to approximate $S_t$ via future rollout loss as done in [Zoran et al., 2021]. However, as we will see in the experiments, this baseline performs poorly in long-term unrolling, since it only considers slots from one timestep.

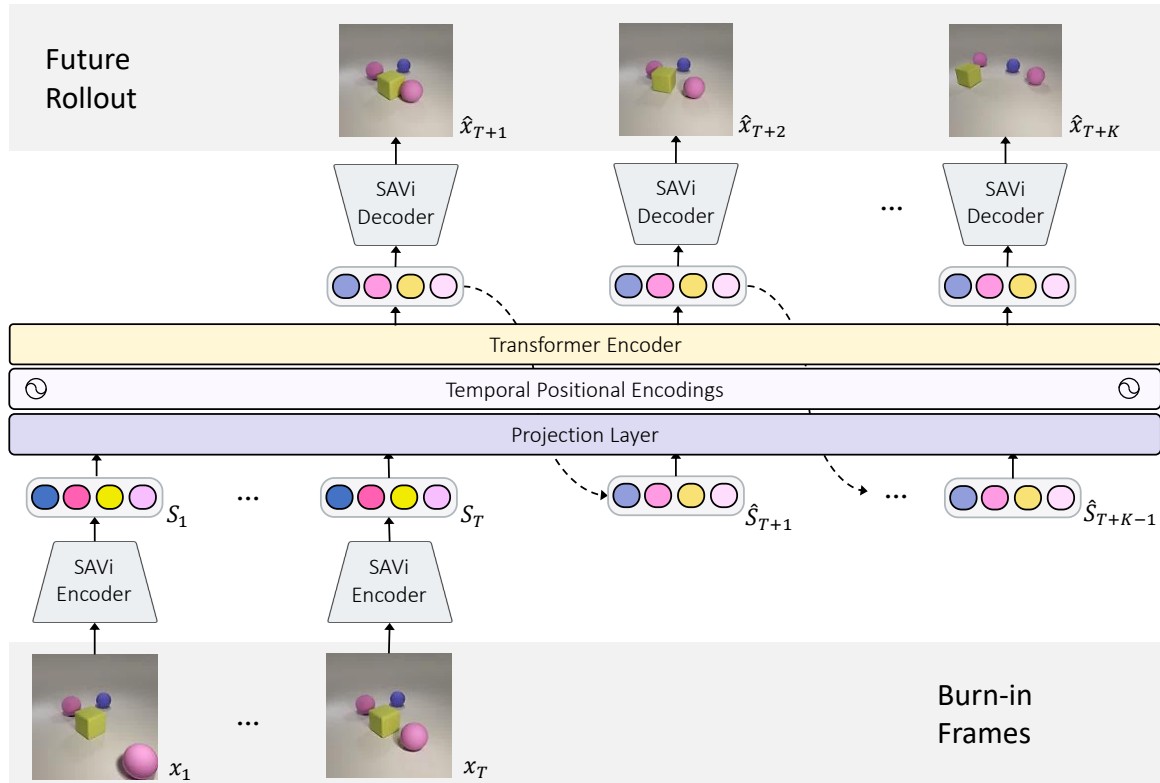

Figure 1: SlotFormer architecture overview. Taking multiple video frames $\{x_i\}_{i=1}^{T}$ as input, we first extract object slots using the pretrained SAVi model. Then, slots are linearly projected and added with temporal positional encoding. The resulting tokens are fed to the model to generate future frames in an autoregressive manner.

## 3.2 SLOTFORMER

**Overview.** Given $T$ input video frames $\{x_i\}_{i=1}^{T}$, SlotFormer synthesizes a sequence of future frames $\{x_{T+k}\}_{k=1}^{K}$ of any given horizon $K$. Our model operates in three steps: i) we convert every video frame into a set of object slots $\mathcal{S}_t$, using a pretrained SAVi encoder, then ii) we feed the slots into a Transformer model that models future dynamics and predict slots at the next time-step, $\hat{\mathcal{S}}_{t+1}$, finally, iii) we feed the predicted slots back into the Transformer to keep generating future rollout autoregressively. To turn the future slots into video frames, we use a pretrained SAVi decoder. The overall pipeline of our method is shown in Figure 1.

**Architecture.** To build the SlotFormer's dynamics model, $\mathcal{T}$, we adopt the standard Transformer encoder module with $N_T$ layers. To match the inner dimensionality of $\mathcal{T}$, we project the input sequence of slots $\{\mathcal{S}_t\}_{t=1}^{T} \in \mathbb{R}^{(TN) \times D_{slot}}$ with linear mapping:

$$G_t = \text{Linear}(\mathcal{S}_t), \qquad (4)$$

where $G_t \in \mathbb{R}^{N \times D_e}$ are the latent embeddings at frame $t$.

To indicate the temporal order of slots, we add positional encoding (P.E.) to the input embeddings. A naive solution would be to add a sinusoidal positional encoding to every slot regardless of its timestep, as done in [Ding et al., 2021a]. However, this would break the *permutation equivariance*

among the slots, which is a useful property of our model. Therefore, we only apply positional encoding at the temporal level, such that the slots at the same time-step receives the same positional encoding:

$$V = [G_1, G_2, ..., G_T] + [P_1, P_2, ..., P_T], \qquad (5)$$

where $V \in \mathbb{R}^{(TN) \times D_e}$ is the resulting input to the transformer $\mathcal{T}$ and $P_t \in \mathbb{R}^{N \times D_e}$ denotes the sinusoidal positional encoding duplicated $N$ times. Not only the temporal positional encoding preserves permutation equivariance among slots, it also gives slightly better prediction results.

Now, we can utilize the Transformer to reason about the dynamics of the scene. Denote the Transformer output features as $U = [U_1, U_2, ..., U_T] \in \mathbb{R}^{(TN) \times D_e}$:

$$U = \mathcal{T}(V). \qquad (6)$$

We use the last $N$ output features, $U_T \in \mathbb{R}^{N \times D_e}$, to predict the slots at the next timestep via a linear mapping:

$$\hat{\mathcal{S}}_{T+1} = \text{Linear}(U_T), \qquad (7)$$

where $\hat{\mathcal{S}}_{T+1} \in \mathbb{R}^{N \times D_{slot}}$.

For consequent future predictions, $\hat{\mathcal{S}}_{T+1}$ will be treated as the ground-truth slots along with $\{\mathcal{S}_t\}_{t=2}^{T}$ to predict $\hat{\mathcal{S}}_{T+2}$. In this way, the Transformer can be applied autoregressively to generate any given number, $K$, of future frames.

*Remark.* The SlotFormer's architecture allows to preserve temporal consistency among slots at different timesteps. To realize the temporal consistency, we employ residual connections from $\mathcal{S}_T$ to $\hat{\mathcal{S}}_{T+1}$, which forces the transformer $\mathcal{T}$ to apply refinement to the slots while preserving their absolute order. Thanks to this useful property, SlotFormer can be used to reason about individual object's dynamics for consistent future rollout.

## 3.3 MODEL TRAINING

Different from previous works that employ a GPT-like architecture which predicts image tokens one by one with a causal attention mask, we directly generate all the slots at the next timestep in parallel. Therefore, we do not need the teacher forcing strategy [Radford et al., 2018] for training. Instead, we train the model using the predicted slots as inputs. This simulates the error accumulation process in long-term sequence generation and improves the quality of the generated videos as will be shown in Section 4.5.

We use a slot reconstruction loss and an image reconstruction loss for training. The former one is the MSE between the predicted and the ground-truth slots, denoted as:

$$\mathcal{L}_S = \frac{1}{K \cdot N} \sum_{k=1}^{K} \sum_{n=1}^{N} ||\hat{s}_{T+k}^n - s_{T+k}^n||^2. \tag{8}$$

We also enforce another MSE loss in the image space to preserve consistent object attributes such as colors and shapes. We obtain the reconstructed image $\hat{x}_{T+k}$ using the pretrained SAVi decoder following (3). The image reconstruction loss is derived as:

$$\mathcal{L}_I = \frac{1}{K} \sum_{k=1}^{K} ||\hat{x}_{T+k} - x_{T+k}||^2. \tag{9}$$

The final objective function is a weighted combination of the two terms with a hyper-parameter $\lambda$:

$$\mathcal{L} = \mathcal{L}_S + \lambda \mathcal{L}_I. \tag{10}$$

## 4 EXPERIMENTS

In this section, we evaluate the future prediction capabilities of SlotFormer on two synthetic video datasets. We aim to answer the following questions:

1. Can an autoregressive Transformer operating on slots generate future frames with high visual quality? (Section 4.2)

2. Does our method achieve consistent long-term dynamics? (Section 4.3)

3. Does a Transformer capture meaningful cues in predicting the future state of objects? (Section 4.4)

4. How does each component of our method contribute to the final performance? (Section 4.5)

## 4.1 EXPERIMENTAL SETUP

**Datasets.** For our evaluation we use *OBJ3D* [Lin et al., 2020] and *CLEVRER* [Yi et al., 2019] - synthetic video datasets capturing objects of diverse appearance, their dynamics and complex interactions, such as collisions and occlusions. OBJ3D consists of videos where a sphere is launched to collide with other objects. We follow [Lin et al., 2020] to use 2,920 videos for training and 200 videos for testing. Since most of the interactions end before 50 steps, we only use the first 50 out of 100 frames in our experiments. CLEVRER is a similar dataset with smaller objects, and there are various objects entering the scene throughout the video, making it harder than OBJ3D. Following [Zoran et al., 2021], we subsample the video by a factor of 2, resulting in a length of 64. We also filter out video clips where there are newly entered objects during the rollout period. Further details on the datasets are provided in Appendix B.

**Implementation Details.** We resize all the images to the $H \times W = 64 \times 64$ resolution, following previous works [Lin et al., 2020, Zoran et al., 2021]. We first pretrain SAVi on OBJ3D and CLEVRER dataset and then extract slots for training SlotFormer. We discovered that vanilla SAVi cannot properly handle some videos on CLEVRER. So we introduce a stochastic SAVi to solve this problem, which will be described in the Appendix. All predictive models are trained by observing $T = 6$ burn-in frames to predict $K = 10$ rollout images. For the Transformer, we use $N_T = 4$ layers, and set $D_e = 128$ on OBJ3D and $D_e = 256$ on CLEVRER. We train our model using a batch size of 64 with Adam optimizer [Kingma and Ba, 2015] on both datasets. The loss weight $\lambda$ is 1. See Appendix C for more details.

**Baselines.** We compare our approach with four baselines which are further described in Appendix D. The first baseline is a naive copy-last-frame method (dubbed *Copy*). We use a video prediction model *PredRNN* [Wang et al., 2017] that directly generates future frames based on global image features as our second baseline. We also adopt the state-of-the-art generative object-centric model *G-SWM* [Lin et al., 2020] which applies sophisticated priors. Finally, since the code of PARTS [Zoran et al., 2021] is not publicly available, we incorporate their Transformer-LSTM based dynamic module to SAVi (denoted as *SAVi-dyn*) and train the model using the same setting as [Zoran et al., 2021].

**Evaluation Metrics.** To evaluate the visual quality of the generated videos, we report PSNR, SSIM [Wang et al., 2004] and LPIPS [Zhang et al., 2018]. As pointed out by [Zhang et al., 2018, Sara et al., 2019], PSNR and SSIM align poorly with human perception, while LPIPS captures more consistent perceptual similarity with human leveraging learned deep features. Therefore, we focus our comparison on LPIPS. To evaluate the predicted object dynamics, we utilize the segmentation mask annotations, and calculate the axis-aligned bounding boxes (AABB) of objects. We

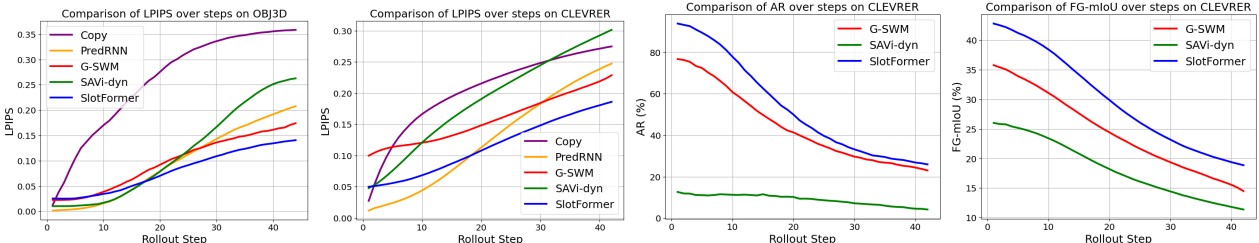

Figure 2: Evaluation of SlotFormer and baselines at each rollout step. We show results in visual quality (left) and object dynamics (right).

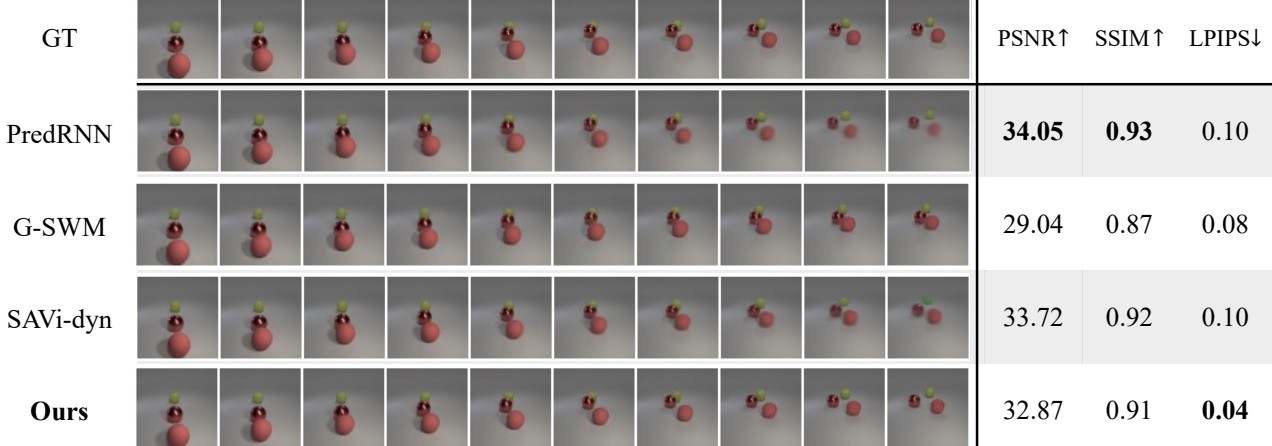

| | | | | PSNR↑ | SSIM↑ | LPIPS↓ |
|---|---|---|---|---|---|---|
| GT | | | | | | |
| PredRNN | | | | **34.05** | **0.93** | 0.10 |
| G-SWM | | | | 29.04 | 0.87 | 0.08 |
| SAVi-dyn | | | | 33.72 | 0.92 | 0.10 |
| **Ours** | | | | 32.87 | 0.91 | **0.04** |

Figure 3: Generation results on OBJ3D. On the right, we report the visual quality metrics of the visualized rollouts for each model.

| Method | OBJ3D | | | CLEVRER | | |
|---|---|---|---|---|---|---|
| | PSNR ↑ | SSIM ↑ | LPIPS ↓ | PSNR ↑ | SSIM ↑ | LPIPS ↓ |
| Copy | 23.14 | 0.77 | 0.26 | 25.56 | 0.84 | 0.20 |
| PredRNN | **33.68** | 0.91 | 0.12 | **31.34** | **0.90** | 0.17 |
| G-SWM | 31.43 | 0.89 | 0.10 | 28.42 | 0.89 | 0.16 |
| SAVi-dyn | 32.94 | **0.91** | 0.12 | 29.77 | 0.89 | 0.19 |
| **Ours** | 32.40 | 0.91 | **0.08** | 30.21 | 0.89 | **0.11** |

Table 1: **Visual quality of the generated frames on both datasets.** Though PSNR and SSIM are poor metrics for this task, we report them for reference. The results of ours are averaged over 3 runs.

| Method | AR ↑ | ARI ↑ | FG-ARI ↑ | FG-mIoU ↑ |
|---|---|---|---|---|
| G-SWM | 43.98 | 57.14 | 49.61 | 24.44 |
| SAVi-dyn | 8.94 | 8.64 | **64.32** | 18.25 |
| **Ours** | **53.14** | **63.45** | 63.00 | **29.81** |

Table 2: **Object dynamics of the generated frames on CLEVRER dataset.** All values are in %. The results of ours are averaged over 3 runs.

calculate the Average Recall (AR) with an IoU threshold of 50% for the predicted object boxes and the Adjusted Rand Index (ARI) for the segmentation masks. We also report a variant of ARI and a variant of mIoU which only focus on foreground objects termed FG-ARI and FG-mIoU as done in [Kipf et al., 2021]. These metrics measure how well our simulated objects follow the ground-truth object trajectories. To test the long-term consistency of different methods, we unroll the model for 44 and 42 frames on OBJ3D and CLEVRER, respectively. By default, all metrics are averaged over the entire rollout horizon.

metrics in this setting. For example, though PredRNN and SAVi-dyn often produce predictions with objects disappearing (as shown in Figure 3), they score highly in these two metrics. In contrast, SlotFormer generates objects with consistent attributes thorough the rollout, which we attribute to modelling dynamics in the object-centric space, rather than in the frames directly. This is also verified in the per-step LPIPS results as shown in Figure 2 left. Since SlotFormer relies on pretrained slots, the reconstructed images at earlier steps have lower quality than baselines. Nevertheless, it achieves clear advantage at longer horizon, demonstrating superior long-term modelling. See Appendix E.1 for more qualitative results on both datasets.

## 4.2 EVALUATION ON VISUAL QUALITY

Table 1 presents the results on visual quality of the generated videos. SlotFormer outperforms all baselines with a sizeable margin in terms of LPIPS, and achieves competitive results on PSNR and SSIM. We note that PSNR and SSIM are poor

## 4.3 EVALUATION ON OBJECT DYNAMICS

To measure how well do the synthesized objects match the ground-truth trajectories, we evaluate the bounding boxes and the segmentation masks of objects. Since OBJ3D does not have such annotations and Copy and PredRNN cannot

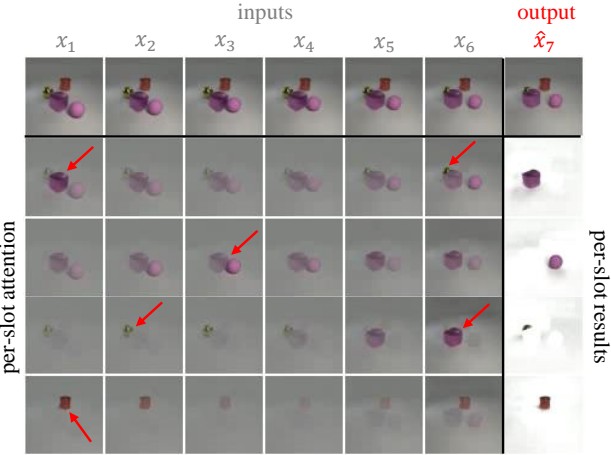

Figure 4: Attention map visualization on the OBJ3D dataset. Our model takes in slots from $\{x_i\}_{i=1}^6$ (column 1-6) to predict slots of $\hat{x}_7$ (column 7). We show images at the first row and the per-slot future reconstructions at the rightmost column. The body of the table shows the per-slot attention of SlotFormer when predicting $\hat{\mathcal{S}}_7$, with the arrows pointing at the regions of high importance for predicting the future slot in the same row.

| Method | PSNR ↑ | SSIM ↑ | LPIPS ↓ |
|---|---|---|---|
| Ours (Full Model) | **32.40** | **0.91** | **0.080** |
|    Burn-in $T = 3$ | 31.26 | 0.88 | 0.093 |
|    Burn-in $T = 4$ | 31.95 | 0.89 | 0.088 |
|    Burn-in $T = 8$ | 32.08 | 0.90 | 0.082 |
|    Naive P.E. | 32.05 | 0.90 | 0.082 |
|    Teacher Forcing | 30.52 | 0.87 | 0.106 |
|    No $\mathcal{L}_I$ | 31.23 | 0.88 | 0.093 |

Table 3: Ablation study on OBJ3D in terms of visual quality.

generate object-level outputs, we do not include them here. Table 2 summarizes the quantitative results. SlotFormer achieves the best performance on AR, ARI and FG-mIoU, and competitive result on FG-ARI. SAVi-dyn scores a high FG-ARI because its blurry predictions assign many background pixels to foreground objects, while the computation of FG-ARI ignores false positives. This is verified by its poor performance in FG-mIoU which penalizes such mistakes. We also show the per-step results in Figure 2 right and Appendix E.2, where our method outperforms other baselines throughout the entire rollout length.

## 4.4 ATTENTION ANALYSIS

In this section, we analyze the visual cues in the past that SlotFormer utilizes to make future predictions. We do so by visualizing the attention map from the last self-attention layer in the transformer $\mathcal{T}$. More precisely, given the last $T$ encoded frames $\{\mathcal{S}_t\}_{t=1}^T$, we are predicting the future slots $\hat{\mathcal{S}}_{T+1}$. Denote the attention scores from $\hat{s}_{T+1}^i$ to $s_t^j$ as $\boldsymbol{a}_{t,j}^i$, where $i, j \in [1, N]$ and $N$ is the number of slots. At each timestep $t$ and for each future slot $i$, we obtain spatial attention maps $o_t^i$ over input frames $x_i$ as a weighted combination of the slot reconstructions as follows:

$$\boldsymbol{o}_t^i = \sum_{j=1}^N \boldsymbol{a}_{t,j}^i \cdot (\boldsymbol{m}_t^j \odot \boldsymbol{y}_t^j), \qquad (11)$$

which indicates the regions of $\boldsymbol{x}_t$ SlotFormer attends upon when predicting $\hat{s}_{T+1}^i$. Figure 4 presents one example, where the purple cube just collided with the purple sphere, and is about to hit the yellow sphere. When predicting the purple cube, the model focuses on the past collision event in $\{\boldsymbol{x}_i\}_{i=1}^4$, and highlights the yellow sphere in $\boldsymbol{x}_6$. For the

purple sphere, the Transformer only looks at the purple cube because it will not hit the yellow sphere. Besides, since the yellow sphere becomes heavily occluded in $\boldsymbol{x}_6$, SlotFormer attends to earlier frames, while predicting its future motion based on the purple cube. Finally, the red cylinder merely looks at itself because it is not involved in the collisions.

## 4.5 ABLATION STUDY

Here, we study the importance of SlotFormer components for future modelling on OBJ3D dataset (see Table 3).

**Burn-in sequence length $T$.** We compare our default burn-in length $T = 6$ to variants with different length. The model performance first improves with more input frames, and slightly drops when $T$ further increases to $8$. We hypothesize that this is because the Transformer is difficult to optimize when the number of input tokens is too large.

**Positional encoding.** Using a vanilla sinusoidal positional encoding which removes the permutation equivariance among slots results small performance drop comparing to the temporal positional encoding. This is not surprising, as permutation equivariance is a useful prior for object modelling, which should be preserved.

**Teacher forcing.** We try the teacher forcing strategy [Radford et al., 2018] by taking in ground-truth slots instead of using the predicted slots autoregressively during training, which degrades the results significantly.

**Image reconstruction loss $\mathcal{L}_I$.** As shown in the table, adding an auxiliary image reconstruction loss improves the quality of the generated videos drastically.

## 5 CONCLUSION

In this paper, we propose SlotFormer to enable object-centric models to perform consistent long-term dynamics modeling. Our approach leverages a Transformer-based autoregressive model to generate plausible future states given a few initial observations of the scene. It leverages the powerful self-attention mechanism to capture spatial-temporal relationship of the scene. Experiments demonstrate that our model can generate videos with high quality and achieves state-of-the-art performance in object dynamics synthesis.

## Author Contributions

- Ziyi Wu and Nikita Dvornik conceived the idea of Transformer-based autoregressive model for object-centric dynamic modeling.

- Ziyi Wu also created the code, conducted the experiments and wrote most of the paper.

- Klaus Greff and Thomas Kipf shared valuable insights in object-centric model implementation and training.

- Nikita Dvornik, Klaus Greff, Thomas Kipf and Animesh Garg provided in-depth advice to the project and paper writing.

- Jiaqi Xi helped with data processing, experiments and figure creation.

## Acknowledgements

We would like to thank Wei Yu for general advice and feedback on the paper, and Xiaoshi Wu, Weize Chen for discussion of Transformer model implementation and training.

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

# A    ADDITIONAL RELATED WORK

We provide a detailed review of related works in this section.

**Dynamics modeling in object-centric representation learning.** SCALOR [Jiang et al., 2019] scales the SQAIR [Kosiorek et al., 2018] model to work on scenes with multiple moving objects. It introduces a background module to model the image background separately. It also equips each object with a depth property to handle occlusions. STOVE [Kossen et al., 2019] incorporates a GNN-based dynamic model into SuPAIR [Stelzner et al., 2019] to reason object interactions, where object representations are explicitly disentangled into positions, velocities and appearance. Similarly, OP3 [Veerapaneni et al., 2020] learns pairwise relationship between objects based on a symmetric assumption. G-SWM [Lin et al., 2020] combines the key properties of the above methods and proposes a unified framework for accurate dynamics prediction. A hierarchical latent modeling technique is utilized to handle the multi-modality of the scene dynamics. Leveraging the power of Transformers, OAT [Creswell et al., 2021] directly learns to align slots extracted from each frame to gain temporal consistency and perform slot interactions. However, the temporal dynamics is still modeled by an LSTM module. Similarly, PARTS [Zoran et al., 2021] employs the same Transformer-LSTM module from OAT. It utilizes the Slot-Attention Locatello et al. [2020] mechanism to detect objects and relies on a fixed independent prior to achieve stable future rollout performance. OCVT [Wu et al., 2021] is the most relevant work to SlotFormer. It also applies Transformer over slots from multiple frames and performs future prediction in an autoregressive manner. However, OCVT still disentangles its underlying object features into position, depth and semantic information. It also relies on a Hungarian matching algorithm to achieve temporal alignment of slots. As a result, OCVT is inferior to G-SWM in terms of future rollout. Compared to previous works, SlotFormer is a general Transformer-based dynamic model that is agnostic to the object-centric representations it builds upon. It does not assume any explicit disentanglement of the object property, while still can handle the object interactions well. Without the use of RNNs or GNNs, we achieve state-of-the-art dynamics modeling ability.

**Transformers.** With the prevalence of Transformers in the NLP field [Vaswani et al., 2017, Kenton and Toutanova, 2019], there have been tremendous efforts in introducing it to computer vision tasks Dosovitskiy et al. [2020], Carion et al. [2020], Liu et al. [2021]. Our method is highly motivated by previous works in Transformer-based autoregressive image and video generation [Esser et al., 2021, Chen et al., 2020a, Yan et al., 2021, Rombach et al., 2021, Ren and Wang, 2022]. VQGAN [Esser et al., 2021] first pretrains the encoder, decoder and a codebook that can map images to discrete tokens and tokens back to images. Then, a GPT-like Transformer model is trained to autoregressively predict the input tokens for high-fidelity image generation. GeoGPT [Rombach et al., 2021] adopts the same image tokens as well as camera tokens as inputs to the Transformer for novel view synthesis. However, their results are not consistent since they only take two views as model input. The design of SlotFormer is mostly related to [Ren and Wang, 2022], which also uses image tokens from multiple frames to enable consistent long-term view synthesis. Different from these works, our mapping step maps images to object-centric representations, preserving the identity of objects.

# B    DATASET DETAILS

Both datasets used in this paper are simulated with physics engine and rendered via Blender [Community, 2018], resulting in physically plausible object interactions such as collision and occlusion and photorealistic image quality.

**OBJ3D.** The objects in this dataset have three shapes (sphere, cylinder, cube), two materials (rubber, metal), three sizes, and five colors, leading to a total of 90 combinations of properties. The videos are generated by first placing 3 to 5 static objects in the scene, and then launching a sphere from the front of the scene to collide with those objects. Compared to CLEVRER, the objects in OBJ3D occupy more pixels in images, have less collisions and occlusions, and are all visible in the scene at the beginning of the videos. Since most of the interactions end before 50 steps, we only train and test the models on the first 50 out of 100 frames.

**CLEVRER.** The objects in this dataset have three shapes (sphere, cylinder, cube), two materials (rubber, metal), one size, and eight colors, leading to a total of 48 combinations of attributes. It is originally designed for Visual Question Answering (VQA) tasks. The videos contain static or moving objects at the beginning, and there will be various new objects entering the scene from random directions throughout the video. The smaller size and more diverse interactions of objects make CLEVRER more difficult than OBJ3D. We obtain the ground-truth segmentation masks from their official website. The bounding boxes are generated from the object masks.

# C    IMPLEMENTATION DETAILS

We provide more implementation details of our method in this section.

**SAVi.** We reproduce the unconditional version of SAVi in PyTorch [Paszke et al., 2019] to perform unsupervised object discovery. Specifically, we use the same CNN encoder, decoder, Slot Attention based corrector and Transformer based predictor as their experiments on CATER. The number of slots $N$ is 6 on OBJ3D and 7 on CLEVRER. The slot size is 128 and the training video clip length is 6 on both datasets. We pretrain SAVi for 80k and 200k steps using the

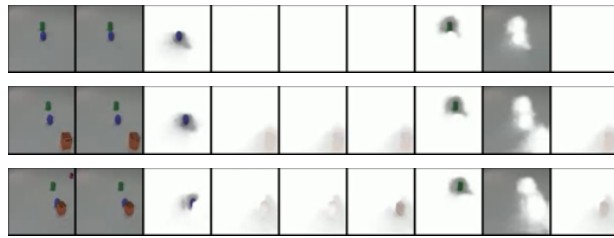

Figure 5: Illustration for missing objects of vanilla SAVi on CLEVRER videos. There are two objects at the beginning of this video (top). When the red cube enters the scene, all 4 empty slots attend to this object, resulting in object sharing (middle). When another object enters the scene from the top right corner, SAVi does not have empty slots to detect it (bottom). As a result, this object is ignored by the model.

Adam [Kingma and Ba, 2015] optimizer with a batch size of 64 on OBJ3D and CLEVRER, respectively. We use the same warmup and decay learning rate schedule which first linearly increases from 0 to $2 \times 10^{-4}$ for the first $2.5\%$ of the total training steps, and then decrease to $0$ in a cosine annealing strategy. We perform gradient clipping with a maximum norm of $0.05$. After pretraining SAVi, we extract slots from each frame and fix them for training our proposed dynamic model.

**Stochastic SAVi.** As stated in the main paper, vanilla SAVi sometimes fails to capture newly entered objects in a video, and we detail the reason and our solution as follows. We use 7 slots for SAVi on CLEVRER which has a maximum of 6 objects in the scene. Imagine a video with 4 objects $O_i{}_{i=1}^4$ at the beginning. Let us assume SAVi captures the objects in the first 4 slots and the background in the 5th slot. This leads to two empty slots $s_6$ and $s_7$, which are very similar with L2 distance $||s_6 - s_7||^2$ generally smaller than $0.05$. Consequently, when there is a new object $O_5$ enters the scene, $s_6$ and $s_7$ will both attend to it, resulting in object sharing between slots. Now, if there is another object $O_6$ entering the scene, there is no empty slot to detect this new object. Therefore, $O_6$ will be ignored by SAVi, until one of the previous object leaves the scene. This issue occurs only on CLEVRER because all the objects are presented in videos of OBJ3D from the beginning, and it happens in scenes with varying number of objects as shown in Figure 5. Besides, SAVi did not experiment on datasets with multiple newly entered objects [1], and thus they did not observe such problem.

From our analysis, the issue stems from the similarity of empty slots, which is because of the permutation equivariance of slots. To break the symmetry, we introduce stochasticity to slots initialized from previous timestep. Specifically, we modify (2) by applying a two-layer MLP with Layer Normalization [Ba et al., 2016] to predict the mean and log

variance of $\tilde{\mathcal{S}}_{t+1}$:

$$(\mu_{t+1}, \log \sigma_{t+1}^2) = \text{MLP}(f_{trans}(\mathcal{S}_t)). \qquad (12)$$

Then, we sample from this distribution to get $\tilde{\mathcal{S}}_{t+1} \sim \mathcal{N}(\mu_{t+1}, \log \sigma_{t+1}^2)$ for performing Slot Attention with visual features at frame $t + 1$.

To enforce this stochasticity, we apply a KL divergence loss on the predicted distribution. Since we do not regularize the mean of $\tilde{\mathcal{S}}_{t+1}$, the loss only penalizes the log variance with a prior value $\hat{\sigma}$:

$$
\begin{aligned}
\mathcal{L}_{KL}^{t+1} &= D_{\text{KL}}(\mathcal{N}(\mu_{t+1}, \log \sigma_{t+1}^2) \,||\, \mathcal{N}(\mu_{t+1}, \log \hat{\sigma}^2)) \\
&= \log \frac{\hat{\sigma}}{\sigma_{t+1}} + \frac{\sigma_{t+1}^2}{2 \cdot \hat{\sigma}^2} - \frac{1}{2},
\end{aligned}
$$
$$(13)$$

which will be averaged over all input timesteps. We set $\hat{\sigma} = 0.1$ which produces enough randomness to break the symmetry without destroying the temporal alignment of slots. With this simple modification, we can detect all the objects throughout the video. We use the same strategy as SAVi to train the stochastic SAVi model on CLEVRER under a combination of the frame reconstruction loss and the KL divergence loss, where the later one is weighted by a factor of $1 \times 10^{-4}$.

**Transformer.** We follow BERT [Kenton and Toutanova, 2019] to implement our model by stacking multiple transformer encoder blocks. The number of self-attention head is 8 and the hidden size of FFN is 512. We adopt the Pre-LN Transformer [Xiong et al., 2020] design as we empirically find it easier to optimize. We train our model using a batch size of 64 for 200k and 500k steps with an Adam optimizer [Kingma and Ba, 2015] on OBJ3D and CLEVRER, respectively. The initial learning rate is $2 \times 10^{-4}$ and decayed to $0$ in a cosine schedule. We also adopt a linear learning rate warmup strategy during the first $5\%$ of training steps. We do not apply gradient clipping or weight decay during training. For all the experiments, we implement our model in PyTorch and train it on 4 NVIDIA RTX6000 GPUs.

# D  BASELINES

We detail our implementation of baselines in this section.

**Copy** is a naive baseline that simply repeat the last frame in the burn-in frames for the entire rollout length. It serves as the lower bound of performance in all the tasks.

**PredRNN [Wang et al., 2017]** is a famous video prediction model leveraging spatial-temporal LSTM to model scene dynamics via global frame-level features. We adopt the online official implementation [2]. The models are trained until convergence for 16 epochs and 6 epochs on OBJ3D

---

[1]confirmed with the authors of SAVi

[2]https://github.com/thuml/predrnn-pytorch

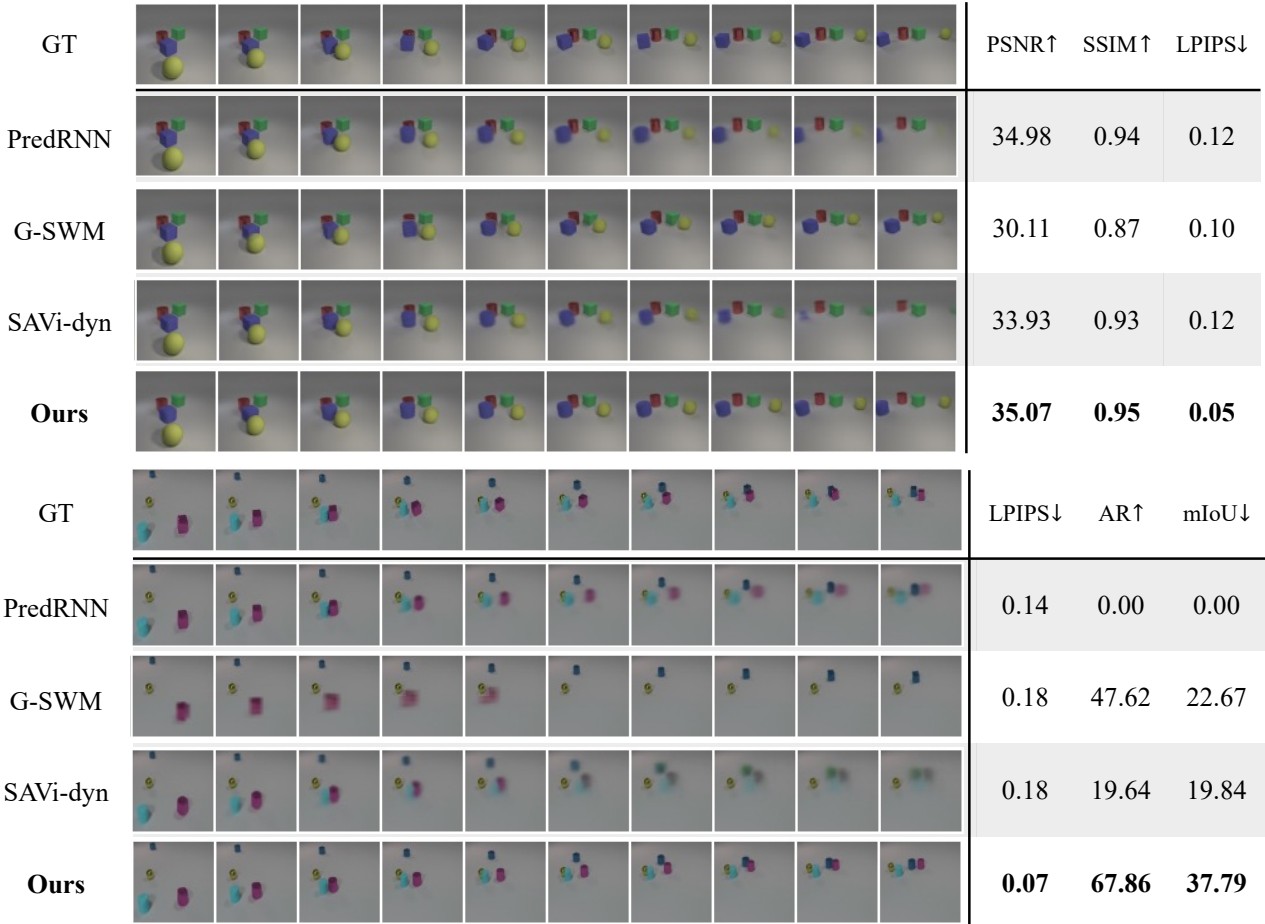

Figure 6: Generation results on OBJ3D (top) and CLEVRER (bottom). On the right, we report the visual quality metrics of the visualized rollouts for each model.

and CLEVRER, respectively. We adopt the same training settings as their original paper.

**G-SWM [Lin et al., 2020]** unifies several priors in previous object-centric models and is shown to achieve good results on various simple video datasets. It constructs a background module to process the scene context, disentangles object features to positional and semantic information, explicitly models occlusion and interaction using depth and GNN module, and performs hierarchical latent modeling to deal with the multi-modality over time. We use the online official implementation [3]. We train the model for 1M steps on both datasets, and select the best weight via the loss on the validation set. Our re-trained model achieves slightly better results than their pretrained weight on the OBJ3D dataset. Therefore, we also adopt the this training setting on CLEVRER.

**SAVi-dyn.** Inspired by the success of PARTS [Zoran et al., 2021], we replace the Transformer predictor in SAVi [Kipf et al., 2021] with the Transformer-LSTM dynamic module in PARTS. The model is trained to observe initial burn-in

frames, and then predict the slots as well as the reconstructed image of the rollout frames using the dynamic module. We use a learning rate of $1 \times 10^{-4}$ and train the model for 500k steps. The other training strategies follow SAVi.

We do not compare with OCVT [Wu et al., 2021] because it underperforms G-SWM even on simple 2D datasets, while SlotFormer outperforms G-SWM under all the settings.

# E MORE EXPERIMENTAL RESULTS

## E.1 QUALITATIVE RESULTS

Figure 6 (top) shows additional qualitative results on OBJ3D. SlotFormer achieves excellent generation of the object trajectories thus very low LPIPS score. However, its PSNR and SSIM are still close to PredRNN and SAVi-dyn, which blurs the moving objects into the background in later frames. This again proves that LPIPS are superior metrics for measuring the generated videos. Besides, G-SWM can also preserve the object identity because it leverages complex priors such as depth to model occlusions. Nevertheless, its simulated dynamics are still worse than our Transformer model.

---

[3]https://github.com/zhixuan-lin/G-SWM

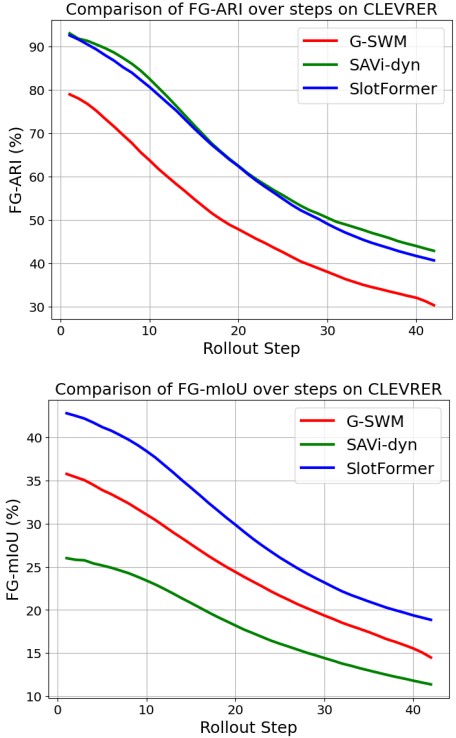

Figure 7: Comparison of the object dynamics of the generated videos at each rollout step on CLEVRER. We report FG-API (left) and FG-mIoU (right) of the segmentation masks.

We present a visual result on CLEVRER in Figure 6 (bottom). The objects are smaller in size and have longer term dynamics, making it much harder than OBJ3D. PredRNN and SAVi-dyn still generate blurry objects at later steps. G-SWM sometimes cannot detect objects newly entering the scene because of the limited capacity of its discovery module. In contrast, SlotFormer builds Transformer on SAVi slots, enabling both accurate object detection and precise dynamics modeling. This is also verified by the object-aware metrics AR and mIoU we show in the figure.

See Figure 9 for more qualitative results.

## E.2 QUANTITATIVE RESULTS ON OBJECT DYNAMICS

We show the per-step FG-ARI and FG-mIoU results in Figure 7. Since SAVi-dyn generates blurry objects, it produces many false positives in the segmentation masks. Instead, SlotFormer preserves the object identity and achieves high scores in both metrics over long rollout steps.

## E.3 ATTENTION MAP ANALYSIS

We show additional attention map visualizations in Figure 8. The top figure presents a scene from OBJ3D dataset, where the blue sphere collides with the red sphere in $x_3$ and will hit the green cylinder soon. SlotFormer looks at the red

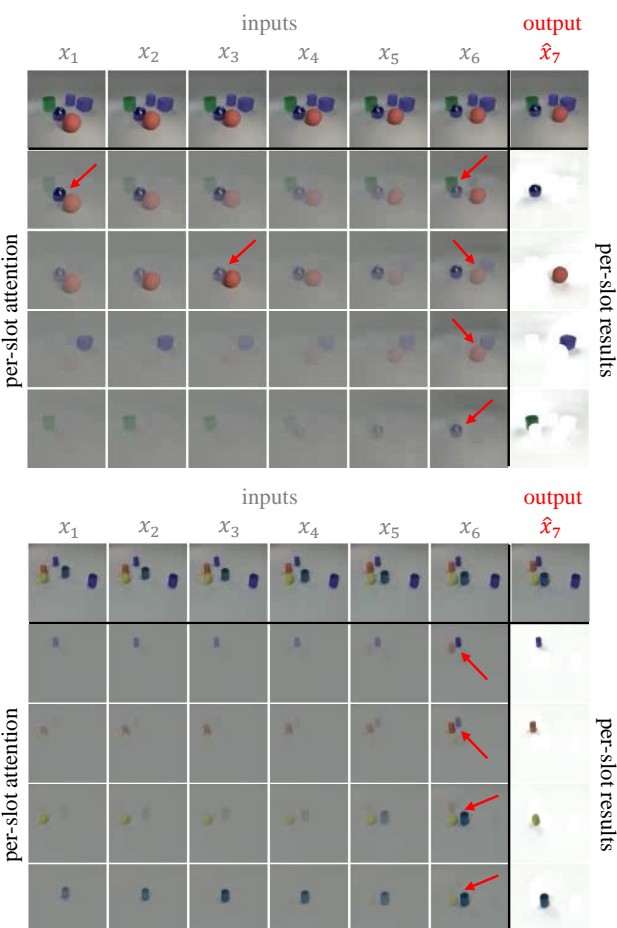

Figure 8: Example attention map visualization on OBJ3D (top) and CLEVRER (bottom). Zoom in for better viewing.

sphere at first and gradually switches to the green cylinder to predict the future blue sphere. For the red sphere, the model focuses on the collision event in earlier frames, while begins to attend on the blue cube it might hit. Similarly, the blue cube and the green cylinder both look at the object that might collide with them.

The bottom one of Figure 8 illustrates one example from CLEVRER. We only analyze the left side of the images since there is no object interaction in the right part. There are two collision events (the purple cylinder hitting the orange cylinder, and the yellow sphere hitting the blue cube) happening in $x_7$, and SlotFormer successfully captures their interactions in the attention maps. In general, we found the attention maps in CLEVRER less clear than those in OBJ3D, due to the smaller object size. Nevertheless, the Transformer can still detect correct object relationships to reason their future motion.

# F   LIMITATIONS AND FUTURE WORKS

**Limitations.** SlotFormer currently builds upon pretrained object-centric models. This family of methods still fail to scale up to real world data, preventing our application to real world videos as well. Besides, the two-stage training strategy harms the model performance at the early rollout steps as shown in Figure 2. It is interesting to explore joint training of SAVi and SlotFormer, which could potentially benefit the performance of both models.

**Future Works.** We only experiment on unconditional generation task in this paper, while we are looking into extending our model to the conditioned video prediction task, such as action-conditioned generation as done in [Zoran et al., 2021]. Some recent works have shown success in this direction by treating conditional inputs as tokens and also feeding them to the Transformer [Ren and Wang, 2022, Rombach et al., 2021, Tevet et al., 2022]. Another direction is the Visual Question Answering (VQA) task which also requires understanding object dynamics [Girdhar and Ramanan, 2019, Bear et al., 2021]. We also want to design other losses for more accurate dynamics modeling, such as the masked embedding prediction loss in [Ding et al., 2021a], and the contrastive loss in [Löwe et al., 2020]. Finally, as uncertainty is necessary for modeling real world dynamics [Lin et al., 2020], we are working on enabling Transformer to represent multi-modality of the future. Overall, we believe SlotFormer is an important step to explore the combination of Transformers and object-centric models, which we see as a promising direction.

| | PSNR↑ | SSIM↑ | LPIPS↓ |
|---|---|---|---|
| GT | | | |
| PredRNN | **32.95** | 0.90 | 0.12 |
| G-SWM | 28.24 | 0.82 | 0.19 |
| SAVi-dyn | 31.75 | 0.89 | 0.13 |
| **Ours** | 31.93 | **0.91** | **0.08** |

| | PSNR↑ | SSIM↑ | LPIPS↓ |
|---|---|---|---|
| GT | | | |
| PredRNN | **33.99** | 0.92 | 0.08 |
| G-SWM | 29.68 | 0.87 | 0.09 |
| SAVi-dyn | 32.65 | 0.92 | 0.10 |
| **Ours** | 31.63 | **0.92** | **0.05** |

| | LPIPS↓ | AR↑ | mIoU↓ |
|---|---|---|---|
| GT | | | |
| PredRNN | 0.11 | 0.00 | 0.00 |
| G-SWM | 0.15 | 43.85 | 25.05 |
| SAVi-dyn | 0.15 | 4.76 | 20.37 |
| **Ours** | **0.04** | **70.04** | **36.73** |

| | LPIPS↓ | AR↑ | mIoU↓ |
|---|---|---|---|
| GT | | | |
| PredRNN | 0.16 | 0.00 | 0.00 |
| G-SWM | 0.20 | 29.76 | 19.81 |
| SAVi-dyn | 0.26 | 11.90 | 8.08 |
| **Ours** | **0.08** | **77.98** | **39.93** |

Figure 9: More qualitative results on OBJ3D (top two) and CLEVRER (bottom two).