# OpenReview forum: "SlotFormer: Long-Term Dynamic Modeling in Object-Centric Models"
_auai.org/UAI/2022/Workshop/CRL — CRL@UAI 2022 Poster_

### Official Review · Reviewer_WK5z · 2022-06-24
**A well written paper with clear contributions**

**Rating:** 7
**Confidence:** 3

**Review:**

**Paper summary:**
This paper introduces a method for long-term object-dynamics modeling. The proposed SlotFormer operates on top of a learned object-centric representation, using a transformer-based autoregressive model to generate future states given T previous states or scene observations. Experiments show that the approach outperforms existing methods on tasks which require long-term synthesis of object dynamics.

**Review summary:**
The ideas in the paper fall under the scope of the workshop, given its focus on structured or object-centric generative models. The paper is well written and presented. The authors did a good job of explaining and empirically verifying their method. The workshop would benefit from having this paper, and the paper would benefit from being at the workshop.

**Pros**
- Clarity of writing
- Quality and rigor of experiments
- Clear contributions

**Cons**
- Limitations and future work not discussed, e.g. does additional flexibility come at the cost of data efficiency? Would provide material for discussions at the workshop.

---

### Official Review · Reviewer_vFgj · 2022-07-01
**good empirical studies**

**Rating:** 6
**Confidence:** 3

**Review:**

This paper proposes a transformer network to model the spatial-temporal dynamics of objects in videos. Given a sequence of images, the proposed SlotFormer takes in the object-centric representation from pretrained SAVi encoder and predicts the object features in future frames. The learning is based on the minimization of reconstruction error of future frames w.r.t. both image and slots.

Strength
The proposed work is well motivated by the drawbacks of existing solutions. The proposed method takes advantage of the power of transformers and model the spatio-temporal dynamics more effectively than previous methods.

The paper is well written and easy to follow. The technical details are sufficient for readers to reproduce the results.

The experimental results are generally better than existing methods in terms of several evaluation metrics.

Weakness
The proposed method is a bit straightforward. The used techniques mostly exist in the literature. The slot representation is given by the SAVi encoder. The proposed slotformer architecture mostly follows the standard transformer architecture.

It is unclear how the proposed method models spatio-temporal dynamics more effectively than previous methods. It seems that the main improvement comes from the transformer architecture, which has been proven to be effective in many sequence tasks. The novel designs to model spatio-temporal dynamics are not totally clear to me.

The proposed method does not show consistent improvement on all evaluation metrics.  It would be better to either point out the importance of different metrics or propose a new metric that can really demonstrate the advantage of the proposed method.

---

### Meta-Review · Program_Chairs · 2022-07-05

**Recommendation:** Accept (Poster)
**Confidence:** 3

**Metareview:**

Both reviewers agree that the proposed method is interesting and well-motivated, and fits perfectly in the workshop. Each of the reviewers proposed some suggestions that might make the paper even better.

---

### Decision · Program_Chairs · 2022-07-06

Accept (Poster)